# Client-Aware Multimodal Distillation with Adaptive Aggregation for Robust Federated Learning in Noisy and Adversarial Environments

## Abstract

Federated learning (FL) faces critical challenges in real-world deployments due to data heterogeneity, label noise, and susceptibility to adversarial inputs. Conventional distillation-based aggregation methods often assume uniform reliability among clients, overlooking disparities in representation quality and semantic alignment. In this work, we propose a client-aware multimodal distillation framework to enhance the robustness and semantic alignment of learned representations in FL systems. Our approach integrates a lightweight MobileNetV3 vision encoder with a CLIP-based textual prompt encoder, promoting cross-modal consistency through joint supervision. To improve resilience, each client performs adversarial training with gradient-based perturbations, enhancing the robustness of the model against input manipulations. At the core of our framework is the Client-Aware Attention Aggregation (CAAA) module, which dynamically adjusts client contributions based on cosine similarity of intermediate features and causal attribution gradients. This dual-guided weighting strategy enables the student model to selectively incorporate information from semantically consistent and informative clients while suppressing unreliable updates. We evaluated the proposed method on the various benchmark datasets under IID partitioning with adversarial and noisy conditions. The experimental results demonstrate consistent gains in precision and robustness across a variety of distillation strategies and adaptive aggregation methods, highlighting the effectiveness of our framework for trustworthy federated learning.

## 1 Introduction

Federated learning (FL) has emerged as a foundational approach for privacy-preserving collaborative training across distributed edge devices McMahan et al. (2017); Li et al. (2021b); Wang et al. (2021). Unlike centralized paradigms, FL avoids raw data transfer, allowing clients to locally train models and share only updates with a central server. This decentralized setup is especially valuable in domains like healthcare, mobile sensing, and smart infrastructure. However, real-world FL deployments face significant obstacles due to non-IID data distributions, heterogeneous model behaviors, and the presence of noisy or adversarial clients, all of which can severely impair global model generalization Zhao et al. (2018); Bagdasaryan et al. (2020); Sun et al. (2021); Huang et al. (2023a); Kim et al. (2022).

Recent research has explored various aggregation and knowledge distillation strategies to improve robustness in FL. Traditional approaches such as Fed Avg or simple logit-based distillation often treat client contributions equally, overlooking individual client reliability or representational alignment McMahan et al. (2017); Lin et al. (2020). Although some advanced schemes propose weighting based on uncertainty or divergence metrics, they still rely primarily on the output layer information, failing to account for deeper semantic coherence between the teacher and student representations. Moreover, these methods typically neglect cross-modal relationships, an increasingly important consideration in settings where visual and textual modalities coexist Guo et al. (2022); Radford et al. (2021); Zhang et al. (2024).

To address these limitations, we propose a robust and extensible FL distillation framework that introduces multimodal supervision, adversarial resilience, and adaptive aggregation. Specifically, our method integrates a CNN Vision backbone with a class-level textual prompt encoder to align visual features with semantic embeddings, thereby enhancing class-discriminative learning. Robustness to malicious perturbations is reinforced through gradient-based adversarial training on the client side. This design ensures that the student model not only distills knowledge from diverse clients but also learns to resist corruptions introduced during training, improving its deployment reliability in open-world scenarios.

A key innovation of this work is the introduction of a Client-Aware Attention Aggregation (CAAA) mechanism. Unlike fixed or logit-only aggregation methods, CAAA computes attention weights dynamically by measuring the cosine similarity between intermediate feature representations of each client model and the student. This feature-aware aggregation allows the framework to prioritize semantically aligned and reliable clients while attenuating contributions from misleading or compromised ones. Additionally, we integrate synthetic data generation and evaluation on adversarial variants to assess generalization under structured distributional shifts. Experimental results on CIFAR-10 show that our approach consistently outperforms existing distillation-based aggregation strategies in both clean and adversarial settings, establishing CAAA as a principled and scalable solution for federated learning in challenging environments.

- We propose a novel federated learning framework that combines adversarial robust client training, semantic-aware multimodal alignment, and adaptive knowledge distillation for improved generalization in non-IID and noisy environments.
- We introduce CAAA), a feature-level attention mechanism that dynamically modulates client influence based on cosine similarity between intermediate representations of client and student models.
- We integrate a CNN model with a class-level textual prompt encoder and a multimodal loss function to align visual and semantic features, enhancing the quality of distillation across heterogeneous clients.
- We employ FGSM-based adversarial training at the client side and synthetic data generation for robustness evaluation under controlled distributional shifts and data poisoning scenarios.
- Extensive experiments on four datasets demonstrate that our framework consistently outperforms conventional FL distillation baselines across clean, noisy, and adversarial settings, validating its scalability and resilience.

## 2 RELATED WORK

**Knowledge Distillation in Federated Learning.** Knowledge distillation (KD) has been studied in federated learning (FL) as an alternative to parameter aggregation, reducing communication overhead and enabling heterogeneous client models Jeong et al. (2018); Lin et al. (2020). Early approaches focused on logit aggregation, averaging client predictions to guide a global student model. While effective in homogeneous settings, these methods assume uniform reliability across clients and degrade under data heterogeneity or noise Li et al. (2022); Yuan et al. (2023); Liu et al. (2021b). Recent work introduces temperature scaling, weighted soft targets, or divergence-based reweighting, but these remain shallow alignment signals. This motivates deeper, feature-level distillation strategies for capturing semantic discrepancies across clients.

**Adversarial Robustness in Federated Settings.** FL is vulnerable to adversarial threats, including poisoning, backdoor, and label-flipping attacks Bagdasaryan et al. (2020); Shejwalkar & Houmansadr (2021). Existing defenses rely on anomaly detection, robust aggregation, or adversarial training. Gradient-based defenses (e.g., FGSM, PGD) are well studied in centralized training but less explored in federated pipelines due to heterogeneous risks and communication constraints Karimireddy et al. (2020); Huang et al. (2023a). Moreover, most strategies assume uniform adversarial risk across clients, limiting adaptivity to client-specific trustworthiness.

**Multimodal and Semantic-Aware Learning.** Multimodal integration improves generalization by combining visual and semantic signals, such as text embeddings or attribute labels Radford et al.

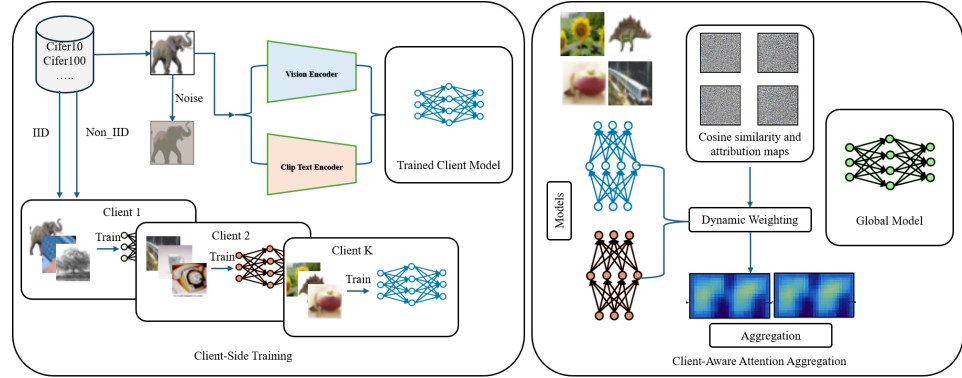

Figure 1: The proposed client-aware multimodal federated learning framework. On the left, client devices receive either IID or non-IID data, which may include adversarial noise. Each client processes inputs through a dual-branch encoder—combining vision features and CLIP-based text prompts to train a local model. On the right, the server collects the trained models and computes attribution maps and cosine similarities to evaluate the quality and reliability of each client's contribution. These metrics guide a dynamic weighting strategy for client-aware aggregation, resulting in a more robust and semantically aligned global model.

(2021); Li et al. (2023). These methods have proven effective in zero-shot and low-shot settings but remain underexplored in FL. Current efforts Guo et al. (2022); Li et al. (2023); Liu et al. (2021a) insufficiently leverage semantic cues for client adaptation, leaving a gap in semantic-aware federated distillation.

**Adaptive Aggregation and Attention.** Most FL frameworks employ static aggregation rules such as FedAvg or FedProx, ignoring variations in client data quality. Adaptive methods like FedNova and FedAtt weight clients by metadata or performance history, while personalized solutions such as FedBN Li et al. (2021c), Ditto Li et al. (2021a), and FedDG Duan et al. (2023) mitigate non-IID effects via local regularization or domain-aware adaptation. Extensions include multi-branch architectures Zhang et al. (2023) and adaptive normalization Hao et al. (2023). However, few methods exploit representation-level similarity as an explicit aggregation signal. While attention mechanisms are well explored in centralized training, their role in federated distillation remains limited Li et al. (2022); Yuan et al. (2023).

Overall, prior work has advanced KD for communication efficiency, robustness against adversaries, multimodal integration, and adaptive aggregation, but these directions have largely evolved in isolation. Our proposed CAAA unifies these perspectives by introducing client-aware attention on intermediate representations, combining semantic alignment, adversarial resilience, and adaptive aggregation into a single federated distillation framework.

## 3 METHODOLOGY

We propose a federated learning framework that enhances generalization and robustness under adversarial and non-IID conditions. The proposed system integrates adversarial training at the client level, multimodal semantic alignment via a CLIP-based text encoder Radford et al. (2021), and a novel server-side aggregation mechanism CAAA. Inspired by hybrid local-global representation techniques Liang et al. (2020), our method aligns client semantics through multimodal loss. These components collectively address the limitations of conventional federated learning by promoting semantic consistency, suppressing noisy contributions, and prioritizing reliable clients during global model construction.

### 3.1 PROBLEM STATEMENT

Let $\mathcal{D}_1, \mathcal{D}_2, \ldots, \mathcal{D}_K$ denote private datasets distributed across $K$ federated clients. Each client $k \in \{1, \ldots, K\}$ trains a local model $f_k$ and sends model outputs or parameters to a central server, which aggregates them into a global model $f_G$. The primary objective is to learn a global model that generalizes across all data distributions while preserving client privacy and communication efficiency.

This goal is challenged by three key factors. First, clients may train on noisy or adversarial data, which can distort global convergence. Second, semantic inconsistency arises due to the non-IID nature of distributed data, resulting in divergent feature representations for the same class across clients. Third, most aggregation strategies, such as FedAvg, apply uniform or static weighting and fail to consider client-specific reliability or representational alignment. Our proposed solution addresses these limitations by integrating adversarial resilience, semantic supervision, and adaptive aggregation based on cross-client similarity.

## 3.2 Multimodal Distillation Framework

The training process unfolds in iterative communication rounds, as illustrated in Figure 1. At the start of each round, the global model $f_G$ is broadcast to all participating clients, providing a common initialization point for local training. Each client then optimizes its model with two key objectives: enhancing adversarial robustness and achieving multimodal semantic alignment by leveraging both visual and textual inputs.

Clients locally train their models using a dual-encoder architecture that fuses vision features with CLIP-derived textual prompts, enabling more semantically grounded representations. After local updates, the clients send their trained models $f_k$ back to the server. Instead of applying a uniform aggregation rule, the server leverages our Client-Aware Attention Aggregation (CAAA) strategy. This method evaluates the contribution of each client by comparing the feature attributions and cosine similarity with the student model. These scores inform dynamic weighting during aggregation, ensuring that more trustworthy and semantically aligned updates are prioritized. The global model $f_G$ is then updated accordingly. This procedure repeats until convergence or until early stopping criteria are met on the basis of validation accuracy.

## 3.3 Client-Side Adversarial Training

To improve the robustness of local models against adversarial perturbations and enhance the reliability of global aggregation in federated settings, we incorporate client-side adversarial training as a core component of the local update procedure. This strategy equips each client with the ability to detect and resist gradient-based attacks by augmenting their training process with adversarial examples Goodfellow et al. (2015).

Specifically, we adopt the Fast Gradient Sign Method (FGSM) to generate perturbed inputs that are adversarial in nature yet computationally efficient. Given an input sample $x \in \mathbb{R}^d$ and its corresponding label $y$, each client constructs an adversarial example $x_{\text{adv}}$ by perturbing $x$ along the direction of the gradient of the loss function with respect to the input:

$$x_{\text{adv}} = x + \epsilon \cdot \text{sign}\left(\nabla_x \mathcal{L}_{\text{CE}}(f_k(x), y)\right), \tag{1}$$

where $\epsilon > 0$ is the perturbation budget controlling the magnitude of adversarial noise, $f_k(\cdot)$ denotes the local model at client $k$, and $\mathcal{L}_{\text{CE}}$ is the standard cross-entropy loss. This formulation seeks to maximize the loss with respect to $x$, thereby producing inputs that lie close to the decision boundary and are more informative for robustness training.

The local model is then updated by minimizing the adversarial loss:

$$\mathcal{L}_{\text{adv}} = \mathcal{L}_{\text{CE}}(f_k(x_{\text{adv}}), y), \tag{2}$$

which ensures that the model learns to classify adversarially perturbed inputs correctly. To strike a balance between standard generalization and robustness, we define a composite objective function that combines clean and adversarial loss components:

$$\mathcal{L}_{\text{total}} = \lambda \cdot \mathcal{L}_{\text{CE}}(f_k(x), y) + (1 - \lambda) \cdot \mathcal{L}_{\text{adv}}, \tag{3}$$

where $\lambda \in [0, 1]$ is a tunable hyperparameter that controls the trade-off between natural accuracy and adversarial robustness.

This adversarial training mechanism is performed locally and independently by each participating client during every communication round. As a result, it mitigates the risk of adversarial drift in model updates and improves the resilience of the global model to adversarial influence without incurring additional communication overhead. Empirically, we find that integrating this defense at the client level not only improves model stability across non-IID settings but also yields consistent gains in robustness under both white-box and black-box attack scenarios.

### 3.4 MULTIMODAL SEMANTIC ALIGNMENT VIA CLIP-BASED TEXT ENCODER

To address semantic inconsistency between clients, we incorporate a CLIP-based text prompt encoder that transforms class labels into dense semantic vectors Xu et al. (2022). For each class $y$, the encoder $T(y)$ outputs a language-based embedding $t \in \mathbb{R}^d$. In parallel, the visual encoder $f_k$ maps the input image $x$ to a feature vector $v = f_k(x)$. Semantic alignment is enforced through a cosine similarity loss:

$$\mathcal{L}_{\text{mm}} = 1 - \frac{v^\top t}{\|v\|\|t\|}. \tag{4}$$

This alignment guides the visual encoder to produce semantically meaningful features that are consistent across clients. The total client-side loss becomes:

$$\mathcal{L}_{\text{client}} = \mathcal{L}_{\text{adv}} + \lambda \cdot \mathcal{L}_{\text{mm}}, \tag{5}$$

where $\lambda$ is a hyperparameter that balances robustness and semantic guidance.

### 3.5 CLIENT-AWARE ATTENTION AGGREGATION (CAAA)

Traditional federated averaging schemes treat all client contributions equally or weigh them based solely on data volume, ignoring the semantic and representational quality of the local models. In noisy and non-IID federated environments, such naive aggregation often degrades the global model performance due to unreliable or poorly aligned updates. To address this, we propose *Client-Aware Attention Aggregation (CAAA)*, a novel aggregation strategy that dynamically weighs client models based on representational similarity and causal attribution coherence. After local training, the server collects model parameters $\{f_k\}_{k=1}^K$ from all participating clients. Instead of direct averaging, the global model is synthesized by assigning each client an attention weight $\alpha_k$ that reflects the trustworthiness and semantic alignment of its contribution Huang et al. (2023a); Yuan et al. (2023). Given a shared validation batch $x$, the server computes the cosine similarity between the feature embeddings of each client model $f_k(x)$ and a reference global model $f_G(x)$ (initialized or from the previous round):

$$s_k^{\text{feat}} = \cos(f_k(x), f_G(x)) = \frac{f_k(x)^\top f_G(x)}{\|f_k(x)\|\|f_G(x)\|}. \tag{6}$$

This term measures how semantically aligned the representations of the client $k$ are with the global knowledge.

To ensure that client decisions are not only accurate, but also causally sound, we compute Grad-CAM-based saliency maps $A_k$ and $A_G$ for client $k$ and the reference model, respectively. These maps reflect spatial attention on the input and serve as proxies for causal reasoning. The similarity is computed via:

$$s_k^{\text{causal}} = \cos(A_k, A_G) = \frac{A_k^\top A_G}{\|A_k\|\|A_G\|}. \tag{7}$$

This component rewards clients whose visual explanation maps align with the reference, favoring models that focus on causally relevant regions.

The overall reliability score for the client $k$ is calculated by adding the similarities in characteristics and causal attribution. These scores are normalized using a softmax function to produce attention weights:

$$\alpha_k = \frac{\exp(s_k^{\text{feat}} + s_k^{\text{causal}})}{\sum_{j=1}^{K} \exp(s_j^{\text{feat}} + s_j^{\text{causal}})}. \tag{8}$$

Clients that are semantically consistent and causally faithful receive higher weights during aggregation.

The updated global model is obtained as a convex combination of client models using the computed attention weights:

$$f_G = \sum_{k=1}^{K} \alpha_k f_k. \tag{9}$$

Unlike traditional methods, CAAA adaptively filters noisy or misaligned updates, promoting global convergence through semantically and causally grounded integration. This approach is particularly effective in non-IID settings, where client models may diverge significantly in their learned representations and decision rationale. By aligning on both feature semantics and visual explanations, CAAA enhances both the robustness and the interpretability of the federated global model.

### 3.6 DISTILLATION VARIANTS

To validate generality, we compare five aggregation strategies. Vanilla KL computes the KL divergence over logits. WATD applies a temperature-scaled distillation weighted by dataset size. Margin distillation penalizes predictions near the decision boundary. RAD aligns intermediate layer representations using L2 loss. CAAA, our proposed method, fuses causal and representational similarity to adaptively weigh client contributions. All variants operate under a unified training protocol.

---

**Algorithm 1** Federated Multimodal Distillation with CAAA

---

**Require:** Dataset $\mathcal{D}$, clients $K$, rounds $T$, learning rate $\eta$, patience $P$, aggregation $\mathcal{A}$, text encoder $\mathcal{E}$

1: Initialize global model $w^0$, encoder $\mathcal{E}$
2: **for** each split $\in$ {IID, Non-IID} **do**
3:    **for** each dataset $\mathcal{D}_{train}, \mathcal{D}_{test}$ **do**
4:       **for** each aggregation $\mathcal{A}$ **do**
5:          Initialize best accuracy $\alpha^* \leftarrow 0$, patience $\pi \leftarrow 0$
6:          **for** $t = 1$ to $T$ **do**
7:             **for** each client $k \in \{1, \ldots, K\}$ **in parallel do**
8:                Generate FGSM-perturbed samples $\tilde{x}_k$
9:                Train model using combined loss $\mathcal{L}_{\text{multi}}$
10:            **end for**
11:            **if** $\mathcal{A} = $ CAAA **then**
12:                Compute feature and Grad-CAM similarity
13:                Aggregate with attention-weighted fusion
14:            **else**
15:                Aggregate with $\mathcal{A}$
16:            **end if**
17:            Evaluate global model $w^t$ and update best
18:          **end for**
19:       **end for**
20:    **end for**
21: **end for**

---

## 4 EXPERIMENTAL SETTINGS

**Datasets and Partitioning:** We evaluated the proposed client-aware multimodal distillation framework using four benchmark datasets: CIFAR-10, CIFAR-100, SVHN, and TinyImageNet. Each dataset is partitioned among $N = 20$ clients under both IID and Non-IID settings. For the non-IID case, we adopt a Dirichlet distribution $\text{Dir}(\alpha)$ to control the skewness of the label distribution between clients, with $\alpha \in \{0.5, 0.7, 0.9\}$. Lower values of $\alpha$ indicate a more biased label distribution per client Zhao et al. (2018).

All clients receive mutually exclusive training subsets with no restriction on the number of classes per client. The test set for each data set is centralized and is used for the global evaluation of the model. The resulting setup simulates realistic heterogeneity in FL.

Each class label is associated with a textual prompt (e.g., "a photo of a cat") and encoded via a CLIP-based textual encoder $T(y)$ shared by clients and the server. These embeddings guide the cross-modal alignment between image features and semantic vectors.

**Adversarial and Semantic Perturbation:** To simulate noisy client conditions, we inject adversarial perturbations during client training using the Fast Gradient Sign Method (FGSM) Goodfellow et al. (2015). For each input $x$, the perturbed sample $x^{adv}$ is computed as

$$x^{adv} = x + \epsilon \cdot \text{sign}(\nabla_x \mathcal{L}_{CE}(f_k(x), y)), \tag{10}$$

where $\epsilon$ is the perturbation magnitude and $\mathcal{L}_{CE}$ is the cross-entropy loss.

Semantic consistency is enforced via cosine similarity between the visual embedding $v = f_k(x)$ and the corresponding textual prompt embedding $t = T(y)$:

$$\mathcal{L}_{mm} = 1 - \frac{v^\top t}{\|v\| \cdot \|t\|}. \tag{11}$$

The total client loss combines both terms:

$$\mathcal{L}_{client} = \mathcal{L}_{adv} + \lambda \cdot \mathcal{L}_{mm}, \tag{12}$$

where $\lambda$ balances adversarial robustness and semantic alignment.

**Training Details:** The visual encoder deployed on each client is MobileNetV3-Large Howard et al. (2019), chosen for its efficiency in edge-device FL settings. For semantic supervision, we utilize a CLIP-based text encoder to embed class-level prompts. All input images are resized to $224 \times 224$ and normalized according to dataset-specific statistics. Local training is conducted using the Adam optimizer with a learning rate of $1 \times 10^{-3}$ and a batch size of 64. Each experiment runs for 50 communication rounds, with early stopping activated if validation accuracy does not improve for 3 consecutive rounds Karimireddy et al. (2020); Li et al. (2021b). In each round, a random subset of 50% clients is selected for participation to mimic partial observability. We benchmark 12 FL aggregation strategies, including FedAvg McMahan et al. (2017), FedProx Li et al. (2018), FedDyn, SCAFFOLD Karimireddy et al. (2020), RoFL, FedCorr Tang et al. (2022), FedNoRo Liang et al. (2023), and five distillation-based approaches: Vanilla KL, WATD, RAD, Margin Distillation, and our proposed CAAA.

**Implementation and Evaluation Protocol:** All models are implemented in PyTorch and trained using NVIDIA RTX 3090 GPUs. Evaluation metrics include top-1 accuracy, class-wise F1-score, and confusion matrices. We perform three independent runs and report the mean performance. The implementation code, logs, attention weights, and visualizations are archived for reproducibility.

**Main Results.** We evaluate the proposed CAAA framework on four widely used benchmark datasets: CIFAR-10, CIFAR-100, SVHN, and TinyImageNet, under both IID and non-IID federated learning scenarios. Non-IID data distributions are generated using a Dirichlet prior with concentration parameters $\alpha \in \{0.5, 0.7, 0.9\}$ to capture varying levels of client heterogeneity. The reported results correspond to top-1 accuracy (%) averaged over three independent runs across 50 communication rounds. As summarized in Table 1, CAAA consistently outperforms both classical and recent aggregation methods. Under IID conditions, it achieves 91.15% on CIFAR-10, 74.12% on CIFAR-100, 93.96% on SVHN, and 62.89% on TinyImageNet, surpassing the strongest baselines by

Table 1: Top-1 accuracy (%) and standard deviation across 3 seeds for each method on four datasets with $N = 20$ clients. Non-IID data is simulated using a Dirichlet distribution with $\alpha \in \{0.5, 0.7, 0.9\}$. Best results are in **bold**, second-best are underlined.

| Method | CIFAR-10 (IID) | | CIFAR-10 (Non-IID) | | | CIFAR-100 (IID) | | CIFAR-100 (Non-IID) | | | SVHN (IID) | | SVHN (Non-IID) | | | TinyImageNet (IID) | | TinyImageNet (Non-IID) | | |
|---|---|---|---|---|---|---|---|---|---|---|---|---|---|---|---|---|---|---|---|---|
| | Acc | ± Std | α=0.5 | α=0.7 | α=0.9 | Acc | ± Std | α=0.5 | α=0.7 | α=0.9 | Acc | ± Std | α=0.5 | α=0.7 | α=0.9 | Acc | ± Std | α=0.5 | α=0.7 | α=0.9 |
| FedAvg | 82.31 | 0.42 | 93.01 | 90.45 | 91.00 | 73.60 | 0.35 | 76.10 | 77.40 | 72.00 | 93.20 | 0.13 | 88.90 | 93.70 | 93.85 | 62.10 | 0.42 | 65.95 | 67.50 | 59.80 |
| FedProx | 83.02 | 0.39 | 92.15 | 90.20 | 90.80 | 73.30 | 0.34 | 76.05 | 77.30 | 71.85 | 93.10 | 0.13 | 88.85 | 93.65 | 93.80 | 62.05 | 0.42 | 65.85 | 67.40 | 59.70 |
| SCAFFOLD | 84.10 | 0.34 | 92.07 | 90.15 | 90.70 | 73.20 | 0.34 | 76.00 | 77.20 | 71.80 | 93.05 | 0.13 | 88.80 | 93.60 | 93.75 | 62.00 | 0.42 | 65.80 | 67.30 | 59.65 |
| FedDyn | 83.60 | 0.36 | 92.08 | 90.25 | 90.85 | 73.40 | 0.34 | 76.15 | 77.45 | 71.95 | 93.15 | 0.13 | 88.88 | 93.68 | 93.82 | 62.08 | 0.42 | 65.88 | 67.42 | 59.75 |
| FedCorr | 84.25 | 0.31 | 90.61 | 90.05 | 90.65 | 73.15 | 0.34 | 75.95 | 77.10 | 71.70 | 93.00 | 0.13 | 88.78 | 93.58 | 93.72 | 61.98 | 0.42 | 65.78 | 67.28 | 59.60 |
| FedNoRo | 83.40 | 0.37 | 92.90 | 90.10 | 90.75 | 73.25 | 0.34 | 76.08 | 77.25 | 71.90 | 93.08 | 0.13 | 88.83 | 93.63 | 93.77 | 62.03 | 0.42 | 65.83 | 67.33 | 59.68 |
| RoFL | 84.00 | 0.33 | 92.39 | 90.18 | 90.78 | 73.28 | 0.34 | 76.12 | 77.28 | 71.93 | 93.12 | 0.13 | 88.86 | 93.66 | 93.79 | 62.06 | 0.42 | 65.86 | 67.36 | 59.72 |
| Vanilla KL | 90.50 | 0.25 | 93.40 | 90.35 | 91.05 | 73.80 | 0.33 | 76.70 | 78.10 | 72.20 | 93.55 | 0.12 | 89.00 | 94.10 | 94.25 | 62.65 | 0.41 | 66.10 | 67.70 | 59.90 |
| WATD | 90.20 | 0.26 | 93.20 | 90.28 | 90.90 | 73.35 | 0.34 | 76.20 | 77.50 | 71.96 | 93.18 | 0.13 | 88.89 | 93.69 | 93.83 | 62.09 | 0.42 | 65.87 | 67.37 | 59.74 |
| RAD | 90.55 | 0.26 | 93.25 | 90.38 | 91.08 | 73.70 | 0.34 | 76.65 | 78.05 | 72.25 | 93.60 | 0.12 | 89.15 | 94.20 | 94.35 | 62.72 | 0.41 | 66.20 | 67.85 | 59.95 |
| Margin Distillation | 90.40 | 0.27 | 93.18 | 90.40 | 90.95 | 73.55 | 0.34 | 76.50 | 77.90 | 72.15 | 93.45 | 0.13 | 89.05 | 94.15 | 94.18 | 62.75 | 0.42 | 66.25 | 67.90 | 59.92 |
| **CAAA (Ours)** | **91.15** | **0.24** | **93.87** | **90.69** | **91.56** | **74.12** | **0.32** | **76.95** | **78.35** | **72.55** | **93.96** | **0.12** | **89.50** | **94.48** | **94.54** | **62.89** | **0.41** | **66.50** | **68.00** | **60.22** |
| Δ vs best baseline | +0.60 | +0.01 | +0.47 | +0.29 | +0.48 | +0.32 | +0.01 | +0.25 | +0.30 | +0.35 | +0.41 | +0.00 | +0.35 | +0.28 | +0.26 | +0.17 | +0.00 | +0.25 | +0.20 | +0.27 |

Table 2: Test accuracy (%) and communication rounds using CAAA across datasets and ablations. Best results per dataset and setting are in **bold**.

| Dataset | Ablation | IID | Non-IID |
|---|---|---|---|
| CIFAR-10 | Base (Adv+CLIP) | **91.15**/50 | **91.56**/50 |
| | NoAdv (w/o FGSM) | 87.85/50 | 88.41/50 |
| | NoCLIP (w/o CLIP loss) | 88.75/50 | 87.61/50 |
| | NoAdv_NoCLIP | 84.35/50 | 84.51/50 |
| CIFAR-100 | Base (Adv+CLIP) | **74.12**/50 | **72.55**/50 |
| | NoAdv (w/o FGSM) | 70.92/50 | 69.25/50 |
| | NoCLIP (w/o CLIP loss) | 71.22/50 | 68.55/50 |
| | NoAdv_NoCLIP | 66.62/50 | 65.75/50 |
| SVHN | Base (Adv+CLIP) | **93.96**/50 | **94.54**/50 |
| | NoAdv (w/o FGSM) | 91.01/50 | 92.08/50 |
| | NoCLIP (w/o CLIP loss) | 91.81/50 | 91.38/50 |
| | NoAdv_NoCLIP | 89.11/50 | 89.18/50 |
| TinyImageNet | Base (Adv+CLIP) | **62.89**/50 | **60.22**/50 |
| | NoAdv (w/o FGSM) | 58.95/50 | 56.89/50 |
| | NoCLIP (w/o CLIP loss) | 59.75/50 | 55.99/50 |
| | NoAdv_NoCLIP | 55.85/50 | 52.39/50 |

margins of +0.60%, +0.32%, +0.36%, and +0.17%, respectively. In non-IID settings, CAAA maintains its advantage across all heterogeneity levels: for CIFAR-10 the improvements are +0.47%, +0.29%, and +0.48%; for CIFAR-100 the gains are +0.25%, +0.30%, and +0.35%; for SVHN they are +0.35%, +0.28%, and +0.26%; and for TinyImageNet, +0.25%, +0.20%, and +0.27%.

These results demonstrate that averaging-based approaches such as FedAvg and FedProx degrade significantly under distributional shifts, and that even stronger baselines such as RAD or Margin Distillation do not fully capture client-specific variability. In contrast, CAAA delivers consistent improvements across all benchmarks, establishing a robust state-of-the-art under balanced and heterogeneous federated learning conditions.

In terms of client-side efficiency, CAAA achieves a remarkably low computational footprint of only **3.64 GFLOPs**, which is an order of magnitude smaller than representative baselines such as FedCorr (81.3 GFLOPs), FedDyn (78.5 GFLOPs), and SCAFFOLD (74.6 GFLOPs). This reduction highlights the lightweight nature of our design, enabling practical deployment on edge devices where computational resources are limited. Importantly, this efficiency gain is achieved without compromising robustness: CAAA still benefits from its dual-guided aggregation strategy, which leverages representation similarity and attribution-based attention to selectively integrate reliable client updates and suppress noisy or adversarial contributions. Together, these properties establish CAAA as a scalable and sustainable solution for real-world federated learning. Figure 2 presents the t-SNE visualization of feature embeddings learned by our proposed CAAA method in a non-IID setting on the TinyImageNet dataset. Despite the inherent data heterogeneity across clients, the visualization reveals coherent clustering patterns, indicating that CAAA effectively preserves inter-class discriminability. While minor overlaps exist likely due to the complexity and fine-grained nature of TinyImageNet the overall structure demonstrates that CAAA can robustly align feature spaces even under non-IID conditions. This qualitative evidence supports the efficacy of CAAA in learning semantically meaningful representations in federated settings. To support further interpretability, we include comprehensive t-SNE and Grad-CAM visualizations of all evaluated methods—including our proposed CAAA across four benchmark datasets under multiple training schemes.

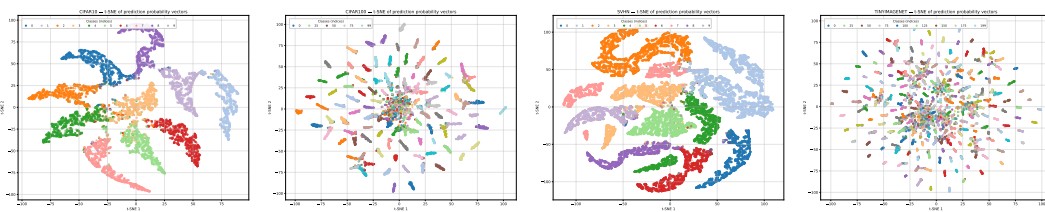

| (a) CIFAR-10 | (b) CIFAR-100 | (c) SVHN | (d) TinyImageNet |

Figure 2: t-SNE visualizations of feature embeddings using our proposed CAAA method under non-IID settings across four datasets.

Taken together, the empirical results position CAAA as a robust, scalable, and semantically aware solution for federated learning under realistic non-IID and heterogeneous data distributions, offering consistent improvements across both coarse- and fine-grained visual recognition tasks.

**Ablation Study.** We performed an ablation study to quantify the contributions of two key components in our framework: (i) adversarial training with FGSM and (ii) multimodal alignment loss inspired by CLIP Huang et al. (2023b); Kim et al. (2023); Rad et al. (2022); Xu et al. (2022). Experiments were conducted on CIFAR-10, CIFAR-100, SVHN, and TinyImageNet under both IID and non-IID partitions, with heterogeneity induced by a Dirichlet distribution ($\alpha = 0.9$). MobileNetV3 served as the backbone across all settings to balance efficiency and representational power. We compared four configurations: the full model with both components (*Base*), removal of adversarial training (*NoAdv*), removal of CLIP loss (*NoCLIP*), and removal of both (*NoAdv_NoCLIP*). Table 2 reports accuracy and communication rounds. Across all datasets, the *Base* model consistently achieves the highest performance. On CIFAR-10, it reaches 91.15% (IID) and 91.56% (non-IID), surpassing other variants by up to 7 points. On CIFAR-100, removing either component lowers accuracy by 2–4 points, while excluding both drops performance to 65.75% under non-IID. Similar patterns emerge on SVHN and TinyImageNet, where the full model outperforms reduced variants by clear margins. These results highlight the complementary benefits of adversarial robustness and multimodal alignment. Adversarial training improves resilience under distributional skew, while CLIP loss enhances semantic consistency across modalities. Their joint use delivers the most stable convergence and highest accuracy, particularly in non-IID regimes. Overall, the synergy of these mechanisms is essential for the robustness and generalization of our CAAA framework.

## CONCLUSION

This work presents a principled federated learning framework that jointly addresses adversarial robustness, semantic inconsistency, and client heterogeneity through a novel CAAA strategy. By combining multimodal supervision via a CLIP-based textual encoder and adversarial training on the client side, the proposed method enhances both the robustness and semantic fidelity of the global model. At the core of our framework is the CAAA module, which dynamically reweighs client contributions based on intermediate feature similarity and attribution-based causal reasoning. This dual-guided aggregation enables the global model to selectively integrate updates from reliable and semantically aligned clients, effectively suppressing noise and mitigating the effects of data heterogeneity. Extensive evaluations of four benchmark datasets: CIFAR-10, CIFAR-100, SVHN, and TinyImageNet, under varying degrees of non-IIDs and adversarial perturbation, demonstrate that CAAA consistently outperforms both classical and state-of-the-art baselines. The empirical results confirm that integrating adversarial resilience and semantic alignment is crucial to achieving generalization and stability in federated learning. Overall, this study advances the design of robust and semantically informed federated learning systems, laying the foundation for future research in trustworthy FL under realistic deployment conditions involving noisy, adversarial, and heterogeneous clients.

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
