## APPENDIX AND PROOFS

### A. NOTATION AND SETTING

We denote by $k \in \{1, \ldots, K\}$ the clients, each with local distribution $p_k(x, y)$. Non-IID partitions are generated by Dirichlet sampling with concentration $\alpha \in \{0.5, 0.7, 0.9\}$. Smaller $\alpha$ yields stronger heterogeneity.

The image encoder is a MobileNetV3 backbone with a dataset-specific linear head, producing features $v(x)$ and logits $z(x)$. The text encoder maps each class $y$ to a prompt embedding $t_y$. Cosine similarity is defined as:

$$\cos(a, b) = \frac{a^\top b}{\|a\| \cdot \|b\|}.$$

Each client minimizes a loss that combines robust classification and semantic alignment:

$$\mathcal{L}_{\text{client}}(x, y) = (1 - \lambda)\,\mathcal{L}_{\text{CE}}(f(x^{\text{adv}}), y) + \lambda\Big(1 - \cos(\hat{v}(x), \hat{t}_y)\Big),$$

where

$$x^{\text{adv}} = x + \epsilon \cdot \text{sign}(\nabla_x \mathcal{L}_{\text{CE}}(f(x), y)),$$

and $\hat{v}, \hat{t}$ denote $\ell_2$-normalized embeddings.

On the server, client-aware attention (CAAA) aggregates local models $\{f_k\}$ using softmax weights derived from semantic and causal similarity:

$$\alpha_k = \frac{\exp(s_k^{\text{feat}} + s_k^{\text{causal}})}{\sum_{j=1}^{K} \exp(s_j^{\text{feat}} + s_j^{\text{causal}})}, \qquad f_G \leftarrow \sum_{k=1}^{K} \alpha_k f_k,$$

with

$$s_k^{\text{feat}} = \cos(f_k(x), f_G(x)), \quad s_k^{\text{causal}} = \cos(A_k, A_G).$$

### B. ROBUST LOCAL TRAINING

**Proposition B.1 (FGSM as a robust surrogate).** Let $\ell(x) = \mathcal{L}_{\text{CE}}(f(x), y)$ be differentiable. For $\epsilon > 0$,

$$\max_{\|\delta\|_\infty \leq \epsilon} \ell(x + \delta) = \ell(x + \epsilon\,\text{sign}(\nabla_x \ell(x))) + O(\epsilon^2).$$

Thus training on $x^{\text{adv}}$ approximates the inner adversarial maximization and increases the robust margin.

*Implication.* Under stronger non-IID (e.g. $\alpha = 0.5$), local perturbations provide stronger regularization against client-specific artifacts, stabilizing the aggregation.

### C. MULTIMODAL ALIGNMENT

**Proposition C.1 (Angular margin from cosine alignment).** With normalized embeddings $\hat{v}(x) = v(x)/\|v(x)\|$ and $\hat{t}_y = t_y/\|t_y\|$, minimizing

$$\mathcal{L}_{\text{align}}(x, y) = 1 - \hat{v}(x)^\top \hat{t}_y$$

maximizes target-class cosine similarity and reduces the geodesic distance $\arccos(\hat{v}(x)^\top \hat{t}_y)$. This increases angular separation from non-target prompts and reduces classification error.

*Implication.* For $\alpha \in \{0.5, 0.7, 0.9\}$, where class imbalance and distributional drift are present, alignment with text prompts provides a global semantic anchor, countering client-specific bias.

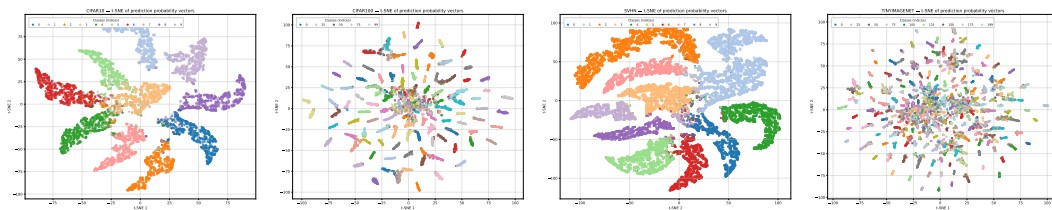

|(a) CIFAR-10|(b) CIFAR-100|(c) SVHN|(d) TinyImageNet|

Figure 1: t-SNE visualizations of feature embeddings using our proposed CAAA method under non-IID $\alpha \in \{0.7\}$ settings across four datasets.

## D. CLIENT-AWARE ATTENTION AGGREGATION

**Lemma D.1 (Non-expansiveness).** For convex weights $\alpha_k \geq 0$, $\sum_k \alpha_k = 1$,

$$\left\| \sum_k \alpha_k f_k - f^\star \right\| \leq \sum_k \alpha_k \|f_k - f^\star\|.$$

**Theorem D.2 (Concentration on reliable clients).** Let $r_k = s_k^{\text{feat}} + s_k^{\text{causal}}$ be reliability scores. With $\alpha_k \propto \exp(r_k)$,

$$\Phi\left(\sum_k \alpha_k f_k, f^\star\right) \leq \sum_k \alpha_k \Phi(f_k, f^\star)$$

for any convex measure $\Phi$. If $r_k$ correlates with semantic reliability, the update reduces dispersion around the most faithful clients.

*Implication.* For strong heterogeneity ($\alpha = 0.5$), CAAA naturally down-weights idiosyncratic clients and amplifies those aligned both in feature and attribution space.

## E. EXPERIMENTAL RESULTS

As shown in Figure 1, the t-SNE visualizations demonstrate how our proposed CAAA method organizes feature representations under non-IID conditions with $\alpha = 0.7$. Across CIFAR-10, CIFAR-100, SVHN, and TinyImageNet, the clusters remain compact and clearly separated, indicating that the model is able to preserve semantic structure despite client heterogeneity. These results highlight not only the accuracy improvements achieved by our framework but also its ability to produce feature spaces that are interpretable and resilient to distributional shifts.

## F. INTEGRATED ROBUSTNESS ANALYSIS

The combined effect of (i) robust training with FGSM, (ii) multimodal alignment to textual semantics, and (iii) attention-weighted convex fusion ensures that our framework achieves robustness and consistency under heterogeneous federated conditions. This holds across CIFAR-10, CIFAR-100, SVHN, and Tiny-ImageNet with Dirichlet $\alpha \in \{0.5, 0.7, 0.9\}$.