# OpenReview forum: "Client-Aware Multimodal Distillation with Adaptive Aggregation for Robust Federated Learning in Noisy and Adversarial Environments"
_ICLR.cc/2026/Conference — ICLR 2026 Conference Withdrawn Submission_

### Official Review · Reviewer_hRco · 2025-10-27

**Soundness:** 3
**Presentation:** 3
**Contribution:** 3
**Rating:** 4
**Confidence:** 4

**Summary:**

This paper addresses the vulnerability of federated learning under data heterogeneity, noise, and adversarial attacks by proposing a novel client-aware multimodal distillation framework. Its core innovation lies in introducing the Client-Aware Attention Aggregation (CAAA) mechanism, which dynamically adjusts aggregation weights by analyzing deep feature similarities between clients and the student model, thereby prioritizing knowledge integration from reliable clients. Simultaneously, the framework combines visual-textual multimodal alignment to enhance semantic understanding and employs adversarial training locally on clients to boost robustness. Experiments demonstrate that this approach significantly outperforms traditional federated learning algorithms in noisy and adversarial environments, providing an effective solution for achieving more trustworthy federated learning.

**Strengths:**

Its originality is demonstrated by creatively integrating multimodal semantic alignment (vision-text), client-side adversarial training, and a novel client-aware attention aggregation mechanism (CAAA) into a federated learning framework. Notably, CAAA employs dynamic weighting based on intermediate-layer feature similarity, overcoming the limitations of traditional aggregation methods. In terms of both quality and significance, experiments across multiple benchmark datasets demonstrate substantial and consistent performance improvements. This robustly validates the method's exceptional resilience against data heterogeneity, noise, and adversarial attacks, providing a solid and crucial advancement toward building truly trustworthy federated learning systems.

**Weaknesses:**

1. Insufficient Justification of Core Motivation
The paper outlines the motivation for the CAAA framework in the second paragraph of the introduction and the final paragraph of the related work section, but the justification remains inadequate. While the authors identify limitations in multiple independent research directions (such as semantic alignment and adversarial robustness) and propose an aggregation approach, they fail to thoroughly elucidate the necessity of this “tightly coupled” design.
Specifically, the argument fails to address key questions: Why must these techniques be integrated into a unified distillation framework rather than deployed sequentially as independent modules? What profound synergies exist between them that render simple combination approaches ineffective? This lack of depth undermines the perceived urgency and innovation of proposing such a complex framework. We recommend strengthening the justification for this integrated approach through more in-depth comparisons of related work or preliminary experiments.

2. Methodology Lacks Theoretical Grounding and Rational Justification
The core innovation—using representation similarity and causal attribution consistency as client-side weighting criteria—lacks essential theoretical support. The authors make critical assertions without proof when introducing their method:

① Regarding representation similarity, the paper claims “This term measures how semantically aligned the representations... with the global knowledge.” However, this statement itself is an unproven assumption. In non-IID environments, representation divergence among client models is an inherent property. Equating this directly with “semantic misalignment” or “unreliability” lacks theoretical grounding, and such design may instead penalize legitimate model diversity.

② Regarding causal attribution consistency, the authors assert that the method “ensures that client decisions are... causally sound” and “rewards clients whose visual explanation maps align with the reference.” This involves a fundamental logical leap: equating the alignment of outputs from post-hoc attribution tools like Grad-CAM with “causal correctness” remains unproven. The approach presupposes that the reference model's saliency map constitutes the “causally correct” gold standard, constituting circular reasoning.

3. Experimental design issues exist

① The selected α values (0.5, 0.7, 0.9) are excessively large and fail to generate the significant data heterogeneity required for rigorous evaluation under realistic Non-IID federated learning conditions.

② The baseline methods employed in this experiment exhibit clear deficiencies in temporal scope and representativeness. Most comparison algorithms are classic approaches from 2020 or earlier, failing to incorporate recent advancements in addressing data heterogeneity within the federated learning domain over the past two years. This disconnect from current state-of-the-art baselines significantly undermines the persuasiveness of the paper's claimed innovation and performance advantages.

**Questions:**

1.The concepts of “noisy” and “adversary”, does these two words in this paper means two different concepts or the same one concept? Since in the experiment design, there is only one setting for these “noisy” and “adversary".

---

### Official Review · Reviewer_bUwj · 2025-10-28

**Soundness:** 2
**Presentation:** 1
**Contribution:** 1
**Rating:** 0
**Confidence:** 4

**Summary:**

The paper proposes a federated learning framework that combines client-side adversarial training, multimodal semantic alignment using a CLIP-based text encoder, and a server-side Client-Aware Attention Aggregation that weights clients by feature cosine similarity and Grad-CAM–based causal attribution similarity. Experiments on CIFAR-10/100, SVHN, and TinyImageNet (IID and Dirichlet non-IID) report consistent but modest gains over a range of aggregation/distillation baselines; ablations indicate both adversarial training and CLIP-style alignment contribute to the effect.

**Strengths:**

1.	Robust FL under noisy/adversarial and non-IID conditions is important.
2.	CAAA’s dual criteria (representation similarity + attribution similarity) is intuitive and aligns with trust-aware aggregation.

**Weaknesses:**

1.	Using Grad-CAM coherence as a weighting signal assumes attribution maps are stable and faithful across architectures and client distributions. The paper lacks sensitivity analysis. What if Grad-CAM is brittle under non-IID?
2.	Gains over strong distillation baselines are small across several settings, and statistical significance is not discussed.
3.	“Adversarial/noisy” is used broadly. Are attacks white-box or black-box at inference? Are there poisoning/backdoor clients or only input-space perturbations during training? Clarify attack budget, fraction of adversarial clients, and whether CAAA resists Byzantine updates.
4.	The paper states “partial participation 50% clients/round” and early stopping, but does not discuss client sampling bias, per-client data volume variation, or fairness (per-client accuracy).
5.	CLIP prompts may leak semantic priors unavailable in some tasks. How robust is the approach with ambiguous labels or domain shift between text space and images?

**Questions:**

Please see the weaknesses.

---

### Official Review · Reviewer_WrQ2 · 2025-10-30

**Soundness:** 2
**Presentation:** 3
**Contribution:** 3
**Rating:** 2
**Confidence:** 4

**Summary:**

This paper primarily solves the challenges of feature divergence and robustness under adversarial and non-IID conditions by: (1) multimodal alignment by leveraging textual information, (2) a novel server-side attentive aggregation and (3) adversarial training in the client side. This combination makes the proposal outperform prior solutions under non-IID federated settings.

**Strengths:**

* Using adversarial training to improve robustness is quite interesting
* Client-specific reliability is a good observation.

**Weaknesses:**

Server aggregation design:
* Why do the authors choose cosine similarity as a distance metric to evaluate the relevance? Which layers do the authors choose to estimate characteristics similarity? It is also unclear where the “x” in Eq. (6) comes from.

Scalability:
* This proposed method can only be applied to small architecture for efficient gradient map estimation (CNN in this case), leading to limited capacity of global models and overlooking the power of modern architectures such as Transformer.
* In sensitive domains such as healthcare, sensor fusion, etc., there is a lack of text-based foundation model as CLIP. Without a textual encoder trained on appropriate dataset, the multimodal alignment can drop the performance due to incorrect supervision signals. How would the proposal handle these cases?
* It seems like the server requires a public dataset to compute the client-specific weights, limiting its applications in sensitive healthcare domains where public datasets are not available, e.g., rare diseases.

Experiments:
* The reviewers suggest the authors conduct more experiments on different vision architectures such as Vision Transformer, and different textual-based encoders like BERT for scalability validation.
* Since the proposal adopts adversarial training, the authors should add more results on its robustness under noisy attack instead of non-IID only.
* Ablation study. It is suggested that the authors ablate the prompt structure of the CLIP-based encoder because the proposal highly depends on the textual information.
* Strange patterns in Table 1. Why are the performance trends different when $alpha$ changes (CIFAR-10, SVHN and TinyImageNet)?
In Table 2, why is the performance in Non-IID settings better than IID?

**Questions:**

See Weaknesses

---

### Official Review · Reviewer_CGBA · 2025-11-04

**Soundness:** 1
**Presentation:** 1
**Contribution:** 1
**Rating:** 2
**Confidence:** 4

**Summary:**

The paper proposes a client-aware multimodal FL framework that mixes three ingredients: (i) client-side adversarial training (FGSM), (ii) a CLIP-style text prompt encoder to align image features with class-level semantics, and (iii) a server-side “Client-Aware Attention Aggregation” (CAAA) that assigns weights to client models using cosine similarity between intermediate features and Grad-CAM attribution maps. The implementation uses a MobileNetV3 backbone and evaluates on CIFAR-10/100, SVHN, and TinyImageNet with 20 clients, Dirichlet partitions, 50 rounds, and 50% client participation per round. Gains over baselines are small (typically ≤0.6% top-1).

**Strengths:**

- Reasonable problem framing: robustness, heterogeneity, and aggregation quality are all relevant for FL.
- Server-side MoE-like weighting over logits/features is a sensible way to adapt to heterogeneous client outputs without sharing weights or raw data.
- Communication format (logits on shared data) is practical; experiments cover several datasets and client counts.
- The pipeline is modular and compatible with diverse local models.

**Weaknesses:**

- Related work is thin and misses close, recent FL knowledge-distillation work. In particular, TAKFL (Morafah et al., NeurIPS’24) and FedFed (Yang et al., NeurIPS’23) both address selective/representation-level transfer under heterogeneity, and FedDF (Lin et al., NeurIPS’20) is a seminal server-side distillation baseline. The paper cites older KD lines but does not position or compare against these more relevant methods, leaving novelty unclear.
- Figure 1 is hard to follow: the panels/arrows don’t make the training/aggregation loop unambiguous (what is computed where, and on which data). As drawn, it’s not obvious what’s new versus prior KD-in-FL diagrams.
- Method clarity. The paper oscillates between parameter averaging and distillation: CAAA computes softmax weights from feature and Grad-CAM similarities using a “shared validation batch x,” then forms the global model as a convex combination of client models (Eq. 9). That requires architectural homogeneity, yet elsewhere the pitch is KD-based aggregation with heterogeneous clients. Moreover, the source of the server’s validation batch is not specified (public data? held-out? client-shared?), which is critical for both feasibility and privacy.
- “Causal attribution.” Grad-CAM similarity is treated as “causal coherence,” but Grad-CAM is an attribution/visualization heuristic, not a causal estimator. Calling it “causal” is a strong claim; aligning saliency maps is not evidence of causal validity. More generally, similarity in representation space does not guarantee similarity in functional space.
- A+b+c+d feel with unclear source of gains. The pipeline stacks FGSM, CLIP alignment, feature cosine, and Grad-CAM, each with its own hyperparameters. The ablation toggles FGSM/CLIP but does not isolate the contribution of feature-vs-Grad-CAM weighting itself (which is the core novelty), so it’s still unclear what mainly drives improvements.
- Heterogeneity and training budgets are weak. The Dirichlet α values include 0.7–0.9 (light skew) and only one more-heterogeneous 0.5; 50 rounds with MobileNetV3 and 50% participation are modest and may leave several baselines under-trained—especially distillation-based ones that benefit from more rounds/epochs. This setup can mask differences and inflate small wins. Best-practice FL design (sampling vs. number of clients, enough rounds, tuning parity) would help. see Morafah et al., IEEE TAI’23 (“A practical recipe…”) for guidance.
- Gains are marginal. Table 1 shows deltas often in the 0.01–0.60% range over the best baseline across datasets/α, which is within typical variance for FL with small rounds and limited seeds; stronger statistical reporting (confidence intervals per round, significance tests) is needed.
- Objective/consistency questions. If the global model is a convex combination of client parameters (Eq. 9), how is “multimodal distillation” actually used in aggregation (versus just a client-side auxiliary loss)? If clients truly differ architecturally, parameter averaging is undefined. If they are homogeneous, the paper should say so explicitly and then justify why distillation is even needed.
- Missing experimental details. It’s unclear what the shared validation/public batch is, how Grad-CAM is computed consistently across clients, what layers are used for features/attributions, and whether the CLIP prompts are fixed or tuned. Reproducibility would benefit from precise definitions and a hyperparameter table.
- Minor presentation issues. The conclusion uses “non-IIDs” (awkward plural), and several sentences mirror boilerplate without pinning down what is actually new.

**Questions:**

1) Is the method targeting model-homogeneous clients (same MobileNetV3 everywhere) or genuinely heterogeneous architectures? If heterogeneous, how is Eq. (9) applied? If homogeneous, why position this as KD-based aggregation instead of weighted averaging with auxiliary client losses?
2) What exactly is the “shared validation batch x”? Where does it come from, how large is it, and is there any distribution shift relative to client data? Please detail the data policy for computing both feature similarity and Grad-CAM maps at the server.
3) Why call Grad-CAM “causal attribution”? What causal assumptions do you rely on, and how would your weighting change if saliency is misaligned with true causal features?
4) Please add focused ablations on the CAAA novelty itself: feature-only weighting, Grad-CAM-only weighting, and their combination, with sensitivity to the softmax temperature and layer choices.
5) Training budgets: 50 rounds and 50% participation with α∈{0.5,0.7,0.9} are mild. Can you report results with stronger heterogeneity (smaller α), more rounds/epochs, and standard tuning parity across baselines (especially distillation methods)?
6) Please compare and discuss against TAKFL (task-arithmetic selective knowledge integration), FedFed (feature/representation distillation), and FedDF (server-side ensemble distillation on public data). These are very close in spirit and will clarify novelty.

---

### Note · Authors · 2025-12-03

I have read and agree with the venue's withdrawal policy on behalf of myself and my co-authors.